# Intrahost SARS-CoV-2 k-mer Identification Method (iSKIM) for Rapid Detection of Mutations of Concern Reveals Emergence of Global Mutation Patterns

**DOI:** 10.3390/v14102128

**Published:** 2022-09-27

**Authors:** Ashley Thommana, Migun Shakya, Jaykumar Gandhi, Christian K. Fung, Patrick S. G. Chain, Irina Maljkovic Berry, Matthew A. Conte

**Affiliations:** 1Viral Diseases Branch, Walter Reed Army Institute of Research, Silver Spring, MD 20910, USA; 2Montgomery Blair High School, Silver Spring, MD 20901, USA; 3Bioscience Division, Los Alamos National Laboratory, Los Alamos, NM 87545, USA; 4Integrated Research Facility, National Institute of Allergy and Infectious Diseases, National Institutes of Health, Frederick, MD 21702, USA

**Keywords:** SARS-CoV-2, COVID-19, variants of concern, intrahost variation, mutation, genomic sequencing, bioinformatics, computational genomics

## Abstract

Despite unprecedented global sequencing and surveillance of SARS-CoV-2, timely identification of the emergence and spread of novel variants of concern (VoCs) remains a challenge. Several million raw genome sequencing runs are now publicly available. We sought to survey these datasets for intrahost variation to study emerging mutations of concern. We developed iSKIM (“intrahost SARS-CoV-2 k-mer identification method”) to relatively quickly and efficiently screen the many SARS-CoV-2 datasets to identify intrahost mutations belonging to lineages of concern. Certain mutations surged in frequency as intrahost minor variants just prior to, or while lineages of concern arose. The Spike N501Y change common to several VoCs was found as a minor variant in 834 samples as early as October 2020. This coincides with the timing of the first detected samples with this mutation in the Alpha/B.1.1.7 and Beta/B.1.351 lineages. Using iSKIM, we also found that Spike L452R was detected as an intrahost minor variant as early as September 2020, prior to the observed rise of the Epsilon/B.1.429/B.1.427 lineages in late 2020. iSKIM rapidly screens for mutations of interest in raw data, prior to genome assembly, and can be used to detect increases in intrahost variants, potentially providing an early indication of novel variant spread.

## 1. Introduction

The unprecedented biomedical research focus on the COVID-19 pandemic has provided an unparalleled amount of genomic data for studying virus evolutionary processes in novel and more detailed ways. Researchers have submitted and made public full-length SARS-CoV-2 genomes together with, and to a lesser extent, the accompanying raw sequencing data in efforts to monitor and surveil how the virus is changing in near real-time [1]. For example, a mutation causing an amino acid change in the Spike protein, D614G, which likely increased the fitness of SARS-CoV-2, spread globally from early to mid-2020 and has since become effectively fixed [2,3]. The amount of change in SARS-CoV-2 genomes remained low until late 2020, when SARS-CoV-2 lineage B.1.1.7 [4], subsequently designated ‘Alpha’ by the World Health Organization (WHO) [5], was identified in the United Kingdom [6]. Alpha exhibited a fitness advantage allowing it to outcompete other circulating lineages [7,8]. The fitness advantage of the Alpha lineage was likely driven by the presence of a number of novel mutations, particularly within the Spike gene where the N501Y change in the receptor binding domain has been shown to increase binding affinity to the ACE-2 receptor [9]. Alpha has also been shown to have a modest ability to evade neutralizing antibodies from prior infection or vaccination [10]. Additional lineages with genetic changes predicted or known to impact spread, disease severity, diagnostic or therapeutic escape, and identified to cause significant community transmission in multiple countries, have been deemed by the WHO as “variants of interest” (VoI) [5]. Such lineages can then be deemed “variants of concern” (VoC) if they also show the ability to cause a detrimental change in COVID-19 epidemiology, increase in virulence, and/or decrease in public health measures [5]. Several additional VoIs and VoCs have been identified: B.1.351/Beta first identified in South Africa [11], P.1/Gamma and P.2/Zeta first identified in Brazil [12,13], B.1.617.2/Delta and AY/Delta first identified in India [14,15], and B.1.1.529/Omicron and BA/Omicron first identified in South Africa and Botswana [16,17].

Most of the genetic sequence analysis of SARS-CoV-2 has focused on consensus genome sequences. However, viruses often exhibit variation within an individual host and exist (and transmit) as a population of variants [18,19]. High throughput genome sequencing methods have been developed to analyze intrahost variation present within genome sequencing data [20,21] and many of the SARS-CoV-2 sequencing experiments are performed using amplicon-based sequencing [22,23]. Intrahost variation in SARS-CoV-2 has now been characterized from several different perspectives including mutation profile differences between intrahost and consensus SNPs [24], specifically within the context of specific geographical regional dynamics [25], across time within the same patients [26,27], and within patients with cancer [28].

Studying intrahost dynamics across hundreds of thousands or millions of samples remains a computationally challenging endeavor both in terms of disk storage of input data and output files, as well as raw compute power. Due to these limitations, previous studies of SARS-CoV-2 intrahost variation have focused on up to ~15,000 datasets [29]. Improving the speed of existing software for analyzing intrahost variation has shown promise [30]. However, alternative approaches for analyzing this large amount of data remain appealing. Counting of relatively short sequences of length k, or ‘k-mers’, has proven to be a very fast and efficient bioinformatics approach for many different types of high throughput sequencing datasets due to the ability to avoid more traditional and time-consuming alignment and post-processing steps [31,32,33]. For instance, a k-mer based tool, fastv, has been developed for detecting SARS-CoV-2 and other pathogens in high throughput sequencing data by providing a set of pathogen-specific k-mers [34]. In addition to providing a SARS-CoV-2 specific k-mer sets, fastv can take as input arbitrary user provided k-mers to allow for flexibility in what a user can screen for. Here, we present iSKIM (“intrahost SARS-CoV-2 k-mer identification method”) as a novel approach with lineage-specific k-mers for the SARS-CoV-2 VoCs/VoIs. These VoC/VoI specific k-mers can then be used for quick k-mer screening of SARS-CoV-2 sequencing datasets to identify samples containing VoC/VoI mutations as intrahost variants and/or consensus variants. iSKIM provides post-processing tools to summarize the screening results and can enable researchers to prioritize particular samples for more complex analyses such as reference-based genome assembly, curation, and downstream analysis when sequencing or analyzing many samples at once.

We also apply iSKIM by scanning for VoC/VoI mutations across publicly available SARS-CoV-2 data in the NCBI Sequence Read Archive (SRA) database. Our analysis identified patterns and trends regarding the frequency of VoC mutations among the datasets and the intrahost diversity of samples. Further application of this technique to newly deposited SARS-CoV-2 raw data may provide an earlier way to forecast potential increases of novel mutations that may become fixed in current or emerging variants.

## 2. Materials and Methods

### 2.1. Variant of Concern Lineage-Specific k-mer Generation

K-mer sequences of 21bp in length were generated for each of the PANGO lineages [4] investigated in this study (B.1.1.7/Alpha, B.1.351/Beta, P.1/Gamma, P.2/Zeta, B.1.429/Epsilon, B.1.526/Iota, and B.1.617.2/Delta). The lineage defining and most common mutations for each lineage were obtained from several sources for validation and completeness [35,36,37]. Typically, lineage defining mutations are listed as amino acid changes in proteins (e.g., Spike N501Y) and do not typically have genome reference coordinates listed. However, to generate k-mers, the nucleotide changes are required. Representative sets of genomes for each lineage were obtained from NCBI and the corresponding lineage defining mutations were matched for those listed in amino acid coordinates to reference coordinates (e.g., Spike N501Y is A23063T). Mutations were defined based on the coordinates of the SARS-CoV-2 reference genome (NCBI accession number NC_045512.2) [38]. Bgzip (version 1.9) and tabix (version 1.9) [39,40] were used to create a compressed and indexed VCF file containing lineage specific mutations, separately for each mutation. These compressed and indexed VCF files were then used to create a consensus reference containing each mutation using the bcftools (version 1.9) ‘consensus’ command by supplying the NC_045512.2 reference and each mutation specific VCF file. A BED file containing the reference coordinate positions 10bp upstream to 11bp downstream of each mutation were created. The bedtools (v.2.30.0) [41] ‘getfasta’ command was then used by supplying the mutation FASTA file previously generated and the appropriate BED file, to generate a 21bp FASTA file for each mutation. Each 21bp k-mer FASTA file was then combined for each lineage to represent the set of mutations for each lineage. Additionally, a set of comparison reference k-mers for each lineage was generated in a similar fashion using simply the SARS-CoV-2 reference sequence (NC_045512.2) at these same positions, but without any mutations.

### 2.2. Obtaining and Formatting NCBI SRA Data

The NCBI SRA database was queried using the phrase “SARS-CoV-2” on 12 May 2021. The BioProject accession identifiers associated with the reads were generated by navigating to the related database section of the page or by querying the BioProject database with “(SARS-CoV-2) AND bioproject_sra[filter] NOT bioproject_gap[filter]”. Only SARS-CoV-2 samples sequenced with the Illumina platform were used for this study. Any samples without the collection month and year, and without a geographic location (country) were not used in the dataset. Any samples with a collection date prior to Nov 2019 were removed. The resulting dataset consisted of 411,805 SRA samples. These NCBI SRA files were downloaded via NCBI FTP [42]. To ensure samples contained SARS-CoV-2 virus genome sequence, only samples with an NCBI “organism” listed as “severe acute respiratory syndrome coronavirus 2” were included as some of these NCBI BioProjects include additional organisms. SRA files were converted to gzip compressed FASTQ files using the ‘fastq-dump’ program from NCBI SRA [43,44] toolkit version 2.10.9 with the following parameters: ‘--split-3 --gzip’.

### 2.3. Screening NCBI SRA Data for Variant of Concern k-mers

Fastv (version 0.8.1) [34] was run on each NCBI SRA fastq.gz file in paired mode (--in1 and --in2) for SRA accessions with paired-end reads, and simply (--in1) for SRA accessions with single-end reads. The custom k-mer sets for each lineage (B.1.1.7, P.1, B.1.351, B.1.429, B.1.617.1, B.1.617.2, and B.1.526.) were supplied separately to fastv with the ‘-k’ option and both html (‘-h’) and JSON (‘-j’) output files were generated.

The fastv JSON output for both the lineage k-mers and corresponding reference k-mer sets were parsed and the proportion of lineage to reference counts were used to determine if mutations belonging to each lineage were present at a minor variant level (>1% to 50%) or as fixed mutations (>99%). This threshold of 1% or greater was chosen to capture a large amount of minor variants without approaching the error rate of various Illumina instruments [45]. A minimum coverage of 5 reads of the reference allele and a minimum of coverage of 5 reads of the mutation allele were required for candidate minor variants.

### 2.4. Inspecting for Primer Induced Mutations Using ARTIC Primer Schemes

The popular ARTIC primer schemes for versions 1 through version 4.1 were downloaded from https://github.com/artic-network/artic-ncov2019/tree/master/primer_schemes/nCoV-2019 accessed on 18 February 2022. The BED files were visualized in IGV [46] alongside specific VoC mutations corresponding to the N501Y and L452R Spike changes to verify that these were not primer induced mutations.

### 2.5. Comparison of iSKIM to LoFreq and ngs_mapper on Select NCBI SRA Data

834 samples containing the N501Y Spike change in samples from October 2020 and the 68 samples containing the L452R Spike change in samples from September 2020 as identified by iSKIM were run through ngs_mapper (version v1.5.4) [47] and LoFreq (version 2.1.4) to compare the k-mer generated call frequencies to frequencies generated by reference-based read assembly. The Wuhan-Hu-1 genome (NCBI accession: NC_045512.2) was used as the reference in both cases.

### 2.6. Phylogenetic Analysis of Select SARS-CoV-2 Genomes

NCBI blastn [48] version 2.11.0+ was used to query the consensus genomes of the 834 NCBI SRA samples identified by iSKIM as having the N501Y Spike change as a minor variant against the GISAID EpiCov database obtained 17 April 2022. The consensus genomes of these 834 samples were downloaded from GISAID. A blastn e-value cut-off (‘-evalue’) of 1e-250 and a percent identity cut-off (‘-perc_identity’) of 99.9 were used. The resulting top 5 blast hits for each query sequence were taken. An additional custom set of 3243 background reference samples were obtained from the NextStrain SARS-CoV-2 global build [49] and added. These were combined with the 834 query sequences and a multiple sequence alignment to the Wuhan-Hu-1 reference (NCBI accession: NC_045512.2) was generated using MAFFT [50] version v7.475 with the following settings: ‘--auto --keeplength --addfragments’. This alignment was used as input to generate a maximum likelihood phylogeny using FastTree version 2.1.11 [51]. The same process was used to generate a tree for the 68 NCBI SRA samples identified by iSKIM as having the L452R Spike change present as a minor variant, except that the top 50 blastn hits were used instead, and all other settings remained the same as described above. Trees were visualized and formatted using FigTree version 1.4.4.

## 3. Results

### 3.1. iSKIM Analysis of SARS-CoV-2 NCBI SRA Data by Month

411,805 samples obtained from the NCBI SRA with collection dates spanning 14 months (February 2020–April 2021) were screened using iSKIM for mutations belonging to the following lineages of concern/interest: B.1.1.7/Alpha, B.1.351/Beta, P.1/Gamma, P.2/Zeta, B.1.429/Epsilon, B.1.526/Iota, and B.1.617.2/Delta. VoC mutations that were either fixed or present as a minor variant in each sample (>1%, see methods for details) were then tabulated across all samples. The spike N501Y change, which is present in most samples of B.1.1.7/Alpha, B.1.351/Beta, and P.1/Gamma, was found in a number of samples either as a fixed variant or as a minor variant (Table 1). Several patterns emerged from examining the N501Y change across each month in this period. 834 samples were detected that had the N501Y change present as a minor variant in October of 2020. The majority of these samples were sequenced and made publicly available in the same month of October 2020 (Appendix A). The number of samples with N501Y present as a minor variant then decreased in November 2020, as the B.1.1.7/Alpha lineage became more prevalent and as the number of samples fixed for N501Y increased. There were 34 samples collected from Australia that had the fixed N501Y change prior to the emergence of B.1.1.7/Alpha or the other VoCs in the June/July of 2020. These samples from Australia have been previously identified in other studies [52,53]. 32 of these Australian samples are assigned to Pango [54] lineage B.1.1.136 and two were assigned to Pango lineage B.1.1 which suggests additional convergence of the N501Y change.

Another key spike change, L452R, has been shown to be associated with increased transmission (in vivo), infectivity (in vivo), and causes reduced antibody neutralization from infected patients and vaccinated individuals [55], as well as escaping HLA-A24-restricted cellular immunity [56]. Spike L452R also shows a similar pattern as N501Y (Table 2). 68 samples were detected that had the L452R change present as a minor variant in September of 2020. All of these 68 samples were sequenced and made publicly available within a month or less of the sample collection date (Appendix A). The number of samples with L452R present as a minor variant then decreased in October 2020 and then in December 2020, three months later, B.1.429/Epsilon became more prevalent and the number of samples fixed for L452R increased.

### 3.2. Phylogenetic Analysis of Early N501Y and L452R Minor Variant Samples

To confirm that the samples containing the N501Y and L452R spike changes identified early as minor variants were not all or primarily found in the same outbreaks or in close transmission chains, global phylogenetic analyses including the consensus genomes of these samples from October 2020 (N501Y) and September 2020 (L452R) were performed. The resulting global trees indicate that the samples containing these minor variant changes emerged independently multiple times (Figure 1 and Figure 2, respectively) and were not part of close transmission chains or related outbreaks. These findings indicate a pattern where a mutation presents itself as a minor variant a few months prior to gaining prevalence as a fixed mutation. The majority of the 834 samples identified as having the N501Y change as a minor variant surge in October 2020 (Table 1) were assigned to Pango lineage B.1.177 and its sublineages (*n* = 477, Appendix A). Similarly, of the 68 samples identified as having L452R as a minor variant surge in September 2020 (Table 2), B.1.177 and its sublineages were the most common although not the majority (*n* = 31, Appendix A).

### 3.3. Comparison of VoC/VoI Mutations

The VoCs/VoIs that were analyzed (B.1.1.7/Alpha, B.1.351/Beta, P.1/Gamma, P.2/Zeta, B.1.429/Epsilon, B.1.526/Iota, and B.1.617.2/Delta) constituted a total of 108 lineage specific mutations, some of which are present in two or more linages (for example, Spike N501Y in Alpha, Beta and Gamma). Of these 108 mutations evaluated with iSKIM between February 2020 and April 2021, *n* = 15 mutations had a substantial rise (*n* > 30 samples) as minor variants prior to fixation (Figure 3 and Appendix A), including the N501Y and L452R Spike changes. *n* = 11 (73.3%) of these mutations were located on either the Spike (*n* = 10) or Nucleocapsid (*n* = 1) structural genes and the remaining *n* = 4 mutations were located on non-structural genes (Table 3). *n* = 42 of the screened VoC mutations had candidate minor variants and fixed variants follow the same growth patterns, where both increase as VoIs/VoCs emerged. Interestingly, *n* = 17 of the mutations had no substantial growth of minor variants despite a rise in the number of fixed variants (Appendix A and summarized in Appendix A). Of those fixed mutations, *n* = 11 (64.7%) were located on non-structural genes and the others were located on Spike (*n* = 3) and Nucleocapsid (*n* = 3).

Of the mutations that were screened for by iSKIM, 50.5% were located on the Spike gene. 36.4% were not located on any of the four structural genes. However, 73.3% of the mutations that peaked as candidate minor variants prior to their fixed variants’ peaks were located on the Spike gene (Table 3) which was significant (one-proportion z-test, *p* = 0.0387). Of the mutations that had no substantial number of samples containing the mutation as minor variants despite a substantial rise in the number of samples containing the mutation as fixed variants, 64.7% of the mutations were not located on structural genes, which was also significant (one-proportion z-test, *p* = 0.0765). Mutations that are located on non-structural genes are more likely to fall into the category of having no substantial rise in minor variant presence paired with a substantial rise in the number of fixed variants.

### 3.4. Comparison of iSKIM to Established Minor Variant Detection Software

To evaluate whether our k-mer based method, iSKIM, produces results that can be repeated by a reference-based assembly method, the samples that iSKIM identified as having the N501Y and L452R changes surge as minor variants (Table 1 and Table 2) were separately run through the variant calling tool LoFreq and ngs_mapper’s built-in variant caller (basecaller.py). Analyzing the 68 samples identified by iSKIM as having L452R as a minor variant in the month of September 2020 (Table 2 and listed in Appendix A), revealed that LoFreq did not identify any of these samples as having sufficient alternate nucleotides to be considered candidate minor variants nor fixed variants for the L452R change, while ngs_mapper identified all 68 samples as having a call frequency between 0 and 0.01. To standardize the comparison between iSKIM and ngs_mapper, the ratio of the T22917G mutation (L452R) to reference was used. The alternate nucleotide (G) to the reference nucleotide (T) [calculated as (# of G)/(# of T)] was compared. The iSKIM ratio tended to be slightly higher than the ngs_mapper ratio (Appendix A). All ratios from both methods were close to 0.01 indicating a low frequency of the mutation in the samples.

834 samples from October 2020 were identified by iSKIM as having N501Y present as a minor variant (Table 1 and listed in Appendix A). LoFreq identified 338 samples as possessing the mutation as a minor variant and 5 as a fixed variant. All 834 samples were run through ngs_mapper, and again, the ngs_mapper output had a lower ratio of mutation to reference nucleotide (A23063T for N501Y) for each sample compared to iSKIM (Appendix A). For the 343 samples that were identified as N501Y variants by all three methods (LoFreq, ngs_mapper, and iSKIM) and registered as either minor or fixed variants, two trends emerged. At higher ratios of mutation to reference nucleotides, when the mutation was fixed or biallelic, iSKIM’s calculated ratio was greater than that of ngs_mapper and LoFreq (Appendix A). However, at lower ratios, when the mutation was a candidate minor variant, iSKIM’s ratio was in between the calculated ratios for ngs_mapper and LoFreq (Appendix A).

The 834 samples from October 2020 with the N501Y change present as a minor variant and 68 samples from September 2020 with the L452R change present as a minor variant were also used to compare the speed of iSKIM with that of LoFreq and ngs_mapper. On average across these samples, iSKIM took 2 min 32 s, ngs_mapper took 1 h 5 min and 28 s, and LoFreq took 1 h 42 min and 31 s to complete (Appendix A).

## 4. Discussion

The COVID-19 pandemic and response has led to an unprecedented amount of genomic data generation and sharing worldwide across publicly available databases. The amount of SARS-CoV-2 genomes and genomic datasets now represents over an order of magnitude greater data than any other previously studied virus [57,58,59]. This includes data across space and time to encompass various waves of the pandemic. In this study we sought to leverage the whole genome sequencing data that is publicly available in the NCBI SRA database to discover sample datasets that contain VoC defining mutations present as intrahost minor variants. Previous studies of SARS-CoV-2 intrahost variation have been performed at smaller scales due to the computational limitations of intrahost analysis [24,25,26,27]. To perform intrahost analysis at a much larger scale we took a different approach by generating short k-mer sequences encompassing VoC mutations that could be used to quickly scan the raw SARS-CoV-2 sequencing reads in the SRA database. Our k-mer based tool, iSKIM, allowed for the scan of over 400,000 raw genomic sequencing datasets totaling dozens of terabytes of raw data.

The analysis of these publicly available data at this scale revealed several patterns. We scanned for SARS-CoV-2 VoC/VoI mutations from the beginning of the pandemic through the emergence of Delta (February 2020–April 2021). 108 total lineage specific mutations were screened and 15 of these mutations had a substantial increase as minor variants in samples detected one to five months prior to fixation. Based on our results, certain mutations appear in the population as minor variants a few months prior to these mutations being seen as fixed mutations in larger numbers of samples. Of the 15 mutations identified with this pattern, 10 were located on the Spike gene, which was statistically significant. Conversely, 17 mutations had no substantial increase in the presence of minor variants despite a rise in the number of samples possessing these mutations as fixed variants. 11 (64.7%) of these mutations were located on non-structural genes of SARS-CoV-2, which was also statistically significant. One possible explanation of this finding is that many of these latter mutations do not confer a fitness advantage to the virus and are neutral mutations that emerged in lineages alongside advantageous mutations that then hitchhike to fixation.

A comparison of iSKIM to LoFreq and ngs_mapper was performed to confirm the accuracy of the iSKIM results. iSKIM consistently called the Spike L452R change at a slightly higher frequency than ngs_mapper, while LoFreq did not call this as a minor variant intrahost mutation in the 68 samples from September 2020 detected by iSKIM. This finding may be explained by the fact that LoFreq employs additional filtering steps that include accounting for strand-bias and high alignment error probability that are not taken into account by the reference-free approach of iSKIM. The minor variant intrahost results of the Spike N501Y change in the 834 samples from October 2020 were comparable across all three methods. In this instance, iSKIM called this intrahost mutation at a slightly higher frequency than ngs_mapper, but at a lower frequency than LoFreq. Therefore, if all 400,000+ NCBI SRA samples analyzed with iSKIM had been analyzed with LoFreq as well, it is possible additional samples containing these VoC mutations may have been identified. However, this is not currently computationally feasible. Nonetheless, the results indicate that iSKIM results correspond well with results from established reference-assembly based methods, ngs_mapper and LoFreq.

Many of the 834 samples from October 2020 that contained the N501Y Spike change as a minor intrahost variant belonged to the B.1.177 Pango lineage, as well as several from the B.1.36.28, B.1.36.17, and B.1.221.1/B.1.221.2 lineages. Each of these lineages have been shown to have been involved in multiple recombination events during the emergence of the Alpha/B.1.1.7 VoC lineage, and none of the recombinant viruses contained a full complement of the Alpha/B.1.1.7 mutations [60]. This pattern is also observed in our analysis of the 834 samples from October 2020, where a subset of the Alpha/B.1.1.7 defining mutations (specifically N501Y), but not all lineage defining mutations, are present as intrahost variants. This may be an important consideration when studying recombination in SARS-CoV-2 [61,62,63].

The large majority of the data analyzed in this study were generated using amplicon sequencing approaches which have been shown to be susceptible to producing varying levels of false primer induced mutations [64]. Primer trimming is a common bioinformatics step employed to remove these artifacts [65]. One shortcoming in the vast amount of SARS-CoV-2 data present in the NCBI SRA is the lack of sufficient metadata and details of the specific primer sets that were used for each run. While many NCBI SRA entries do include the sequencing strategy details that were used, for example, typical ARTIC protocols [22], these primer sets are updated regularly and primer sequences are not included with the sequencing submission. Therefore, it was not feasible in this study to primer trim each of the 400,000+ samples that were analyzed. However, for the main findings of the L452R and N501Y changes, the popular ARTIC primer schemes were considered during our analyses as the 834 and 68 samples identified were generated with the ARTIC protocol. Neither of the mutations (T22917G/L452R or A23063T/N501Y) overlapped with ARTIC primers (see Methods). Therefore, these two intrahost mutations that we have highlighted were not impacted by primer induced mutations in the samples identified.

In this study we focused on SARS-CoV-2 Illumina sequencing data that was available in the NCBI SRA as this represented a very large amount of data. There is also a large amount of Oxford Nanopore SARS-CoV-2 sequencing data as well as other platforms such as PacBio. The error profiles of these longer read technologies differ from that of Illumina. Incorporating iSKIM support for these other data with varying error profiles should entail adjusting several settings within iSKIM to account for differences between reads and k-mers while also providing sufficient sensitivity for detection. Similarly, this may also provide a way to account for k-mer erosion if additional SARS-CoV-2 mutations accumulate within the chosen k-mer sequences.

In addition to screening the large number of raw samples publicly available in the NCBI SRA, iSKIM can also be used to rapidly screen for newly sequenced samples that contain VoC or other mutations of interest prior to the more time-consuming and computationally expensive steps of reference-based genome assembly and curation. This can allow researchers to prioritize particular samples for reference-based genome assembly, other downstream analyses, or early reporting when turnaround time is critical. This has been particularly useful during periods of the pandemic when new VoCs are emerging, but not yet taken over as the dominant variant circulating. The iSKIM results also provide a complementary view of the data alongside typical consensus genome results.

Important putative and known mutations in the SARS-CoV-2 genome have been identified that may allow the virus to escape immune defenses [66,67,68,69]. Studies of patients with various forms of immunosuppression have revealed divergent SARS-CoV-2 virus sequences [70,71,72]. Additional mutations rarely observed in genome sequences sampled from clinical settings have been found in abundance in certain wastewater surveillance [73]. Animal reservoirs also pose a potential source of additional SARS-CoV-2 variation with the capability for spillback into humans [74,75]. One additional way in which iSKIM could be applied would be to manually gather and curate this growing list of SARS-CoV-2 mutations not seen in previous or currently circulating VoC lineages. These mutations would then be added as sets of k-mers to iSKIM and could be used to screen all newly submitted raw sequencing datasets as a way to provide an early warning that known mutations may be emerging first as minor variant mutations. It still may be difficult to discern which of these mutations may be more important to pursue experimentally and which are less critical. However, this approach could provide a slightly earlier detection to when many of these mutations are soon then seen at the consensus level as is the current paradigm for early detection and warning.

## Figures and Tables

**Figure 1 viruses-14-02128-f001:**
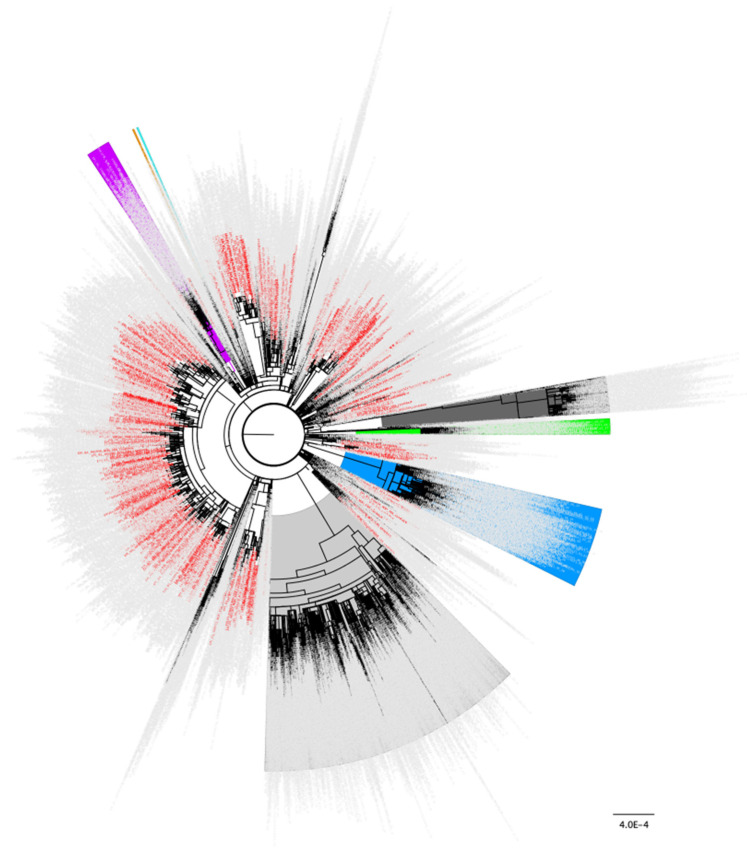
Distribution of the 834 samples containing Spike N501Y as a minor variant from October 2020 across the global SARS-CoV-2 phylogeny indicating independent emergence. Background lineages include Alpha/B.1.17 samples highlighted in blue, Gamma/P.1 highlighted in green, Beta/B.1.351 highlighted in purple, Epsilon/B.1.429 highlighted in orange, Iota/B.1.526 highlighted in turquoise, Delta/B.1.617.2 highlighted in grey, and Omicron/BA.1/BA.2 highlighted in dark grey. None of the 834 samples containing Spike N501Y as a minor variant in October 2020 were present in these highlighted lineages. Non VoC/VoI lineages are not highlighted. 834 samples identified as having the N501Y change present as a minor variant in October 2020 (Table 1) are colored in red. 3243 total background genomes were included in this analysis.

**Figure 2 viruses-14-02128-f002:**
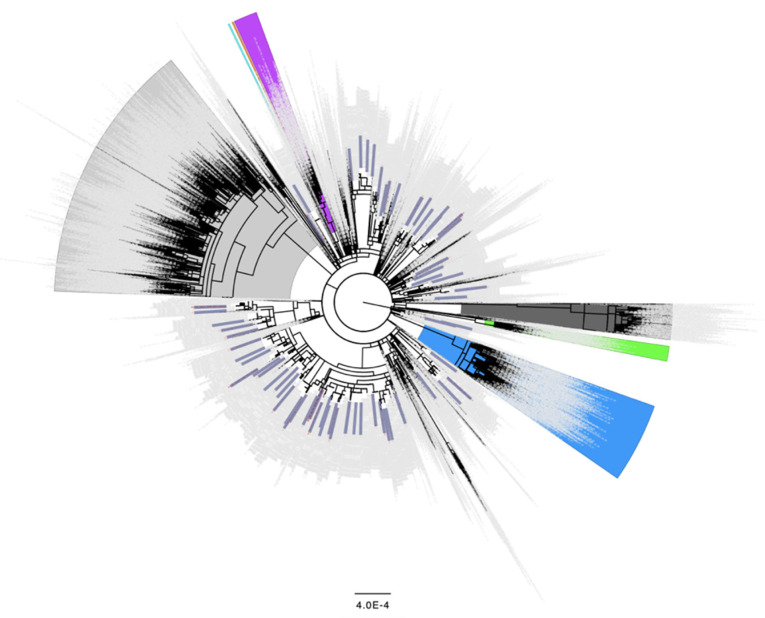
Distribution of the 68 samples containing Spike L452R as a minor variant from September 2020 across the global SARS-CoV-2 phylogeny indicating independent emergence. Background lineages include Alpha/B.1.17 samples highlighted in blue, Gamma/P.1 highlighted in green, Beta/B.1.351 highlighted in purple, Epsilon/B.1.429 highlighted in orange, Iota/B.1.526 highlighted in turquoise, Delta/B.1.617.2 highlighted in grey, and Omicron/BA.1/BA.2 highlighted in dark grey. None of the 68 samples containing Spike L452R as a minor variant in September 2020 were present in these highlighted lineages. Non VoC/VoI lineages are not highlighted. 68 samples identified as having the L452R change present as a minor variant in September 2020 (Table 2) are colored in blue rectangles. 3243 background genomes were included in this analysis.

**Figure 3 viruses-14-02128-f003:**
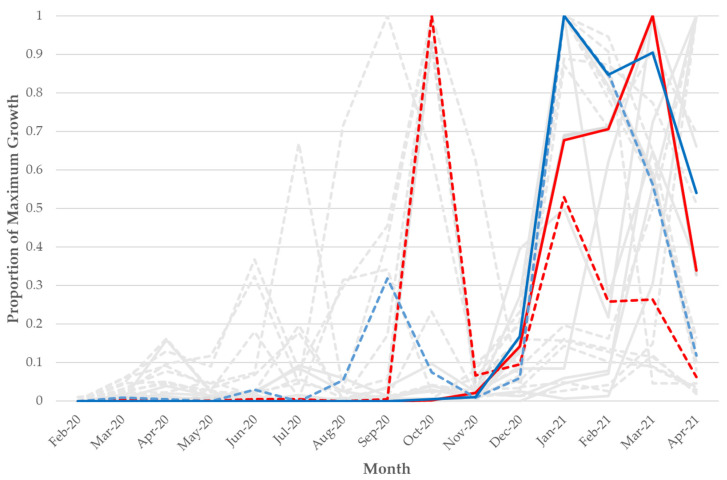
Frequency over time of the *n* = 15 VoC/VoI mutations that had a substantial increase as a minor variant prior to a rise as a fixed variant across 411,805 NCBI SRA SARS-CoV-2 samples. The Y-axis is scaled by the maximum count for each particular mutation and scaled separately for minor variant and fixed mutations. Dotted lines represent minor variant mutations and solid lines represent fixed mutations. The red solid and dotted lines represent the A23063T/N501Y mutation/change, and the blue solid and dotted lines represent the T22917G/L452R mutation/change. The grey lines represent the other 13 VoC/VoI mutations that had a substantial increase as a minor variant prior to a rise as a fixed variant (each is also found in Appendix A).

**Table 1 viruses-14-02128-t001:** Number of samples containing the Spike N501Y change present as a fixed variant or minor variant in NCBI SRA samples across each month as identified by iSKIM. Numbers in bold represent months where a high frequency of samples with the N501Y change was identified prior to emergence first in the Alpha VoC.

Month and Year	# of NCBI SRA Samples Screened	# NCBI SRA Samples Fixed for N501Y	Fraction of NCBI SRA Samples Fixed for N501Y	# of Samples with N501Y Present as a Minor Variant	Fraction of Samples with N501Y Present as a Minor Variant
February 2020	298	0	0.0000	0	0.0000
March 2020	14,279	0	0.0000	3	0.0002
April 2020	16,396	0	0.0000	2	0.0001
May 2020	8085	0	0.0000	1	0.0001
June 2020	10,381	**31**	0.0030	4	0.0004
July 2020	10,344	**3**	0.0003	5	0.0005
August 2020	9646	0	0.0000	0	0.0000
September 2020	11,000	19	0.0017	5	0.0005
October 2020	22,710	240	0.0106	**834**	**0.0367**
November 2020	22,671	1618	0.0714	56	0.0025
December 2020	26,274	10,405	0.3960	80	0.0030
January 2021	69,019	49,666	0.7196	442	0.0064
February 2021	61,025	51,801	0.8488	216	0.0035
March 2021	81,301	73,298	0.9016	220	0.0027
April 2021	28,507	24,882	0.8728	53	0.0019

**Table 2 viruses-14-02128-t002:** Number of samples containing the Spike L452R change present as a fixed variant or minor variant in NCBI SRA samples across each month as identified by iSKIM. Numbers in bold represent the month where a high frequency of samples with the L452R change was identified prior to emergence first in the Epsilon VoI.

Month and Year	# Of NCBI SRA Samples Screened	# NCBI SRA Samples Fixed for L452R	Fraction of NCBI SRA Samples Fixed for L452R	# Of Samples with L452R Present as a Minor Variant	Fraction of Samples with L452R Present as a Minor Variant
February 2020	298	0	0.0000	0	0.0000
March 2020	14,279	0	0.0000	2	0.0001
April 2020	16,396	0	0.0000	1	0.0001
May 2020	8085	0	0.0000	0	0.0000
June 2020	10,381	0	0.0000	7	0.0007
July 2020	10,344	0	0.0000	0	0.0000
August 2020	9646	0	0.0000	11	0.0011
September 2020	11,000	0	0.0000	**68**	**0.0062**
October 2020	22,710	8	0.0004	15	0.0007
November 2020	22,671	17	0.0007	2	0.0001
December 2020	26,274	257	0.0098	11	0.0004
January 2021	69,019	1525	0.0221	201	0.0029
February 2021	61,025	1293	0.0212	172	0.0028
March 2021	81,301	1381	0.0170	113	0.0014
April 2021	28,507	825	0.0289	23	0.0008

**Table 3 viruses-14-02128-t003:** *n* = 15 VoC/VoI mutations that appeared as candidate minor variants prior to becoming fixed variants were mostly associated with the spike protein including on the NTD and RBD protein domains. ‘X’ denotes which lineage(s) each mutation is predominantly found in.

Mutations of Concern (Amino Acid Notation)	Protein Segment	Lineages					
		P.1/Gamma	B.1.1.7/Alpha	B.1.351/Beta	B.1.429/Epsilon	B.1.617.1/Iota	B.1.617.2/Delta
ORF1a: G5230T (K1655N)	NSP3			X			
ORF1ab: G17014T (D260Y)	NSP13				X		
ORF1ab: G17523T (M1352I)	NSP13					X	
Spike: G21600T (S13I)	NTD				X		
Spike: G21974T (D138Y)	NTD	X					
Spike: G22132T (R190S)	NTD	X					
Spike: T22917G (L452R)	RBD				X	X	X
Spike: G23012C (E484Q)	RBD					X	
Spike: A23063T (N501Y)	RBD	X	X	X			
Spike: C23271A (A570D)	CTD		X				
Spike: C23604A (P681H)	CTD		X				
Spike: T24506G (S982A)	CTD		X				
Spike: G24914C (D1118H)	CTD		X				
Nucleocapsid: G28881T (R203M)	Nucleocapsid					X	X
ORF8: G28048T (R52I)	ORF8		X				

## Data Availability

Not applicable.

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
