# Peer review of "Intrahost SARS-CoV-2 k-mer Identification Method (iSKIM) for Rapid Detection of Mutations of Concern Reveals Emergence of Global Mutation Patterns"

_viruses, 2022, doi:10.3390/v14102128_

Round 1

Reviewer 1 Report

The manuscript by Thommana et al. describes a software platform developed for rapidly screening viral lineages using a k-mer approach to identify and quantify viral sub-variants of concern.  They used this software program to analyze the emergence of individual PMs associated with Alpha and Epsilon lineages of SARS-CoV-2.  Surprisingly, they found that there were increases patient sequences with subvariants containing individual PMs associated with the VOCs several months before the emergence of the VOCs themselves.  While I don't have any problems with the methods themselves, I am very suspicious of the conclusions.  The data output can only be as good as the data itself and I have seen so many instances of sequencing and entry mistakes associated with SARS-CoV-2 genomes.  I am not convinced that there isn't a technical explanation for their observation (since it makes no sense to me biologically), but that is not the fault of their analysis.

Major issues.

1. The major point of the software as I understand it is that it is much faster and requires less computing power.   Therefore, I really think it would benefit the manuscript to quantify and display this information more clearly and directly.  How many seconds/minutes/hours does it take to run a X number of samples using the different software platforms?  How much better/faster is it?

2. As I was reading the manuscript my major concern was that the main conclusion was a result of low level contamination or barcode misread from the submitting lab.  Even though the early spike in N501Y sub-variants was seen from samples collected in October 2020, we have no way of knowing when the samples were actually sequenced.  If it were not sequenced for a few months, it might have been run along with a lot of samples that were 100% N501Y, and the possibility of the sequence coming from some form of cross-contamination goes up.  My concern about this was largely (but not completely) alleviated by the information in Supp Fig 1 that showed that the different VOC PMs did not appear all together.  I really think it would be useful if the authors showed this data a little bit more upfront.  An easy solution would be to include the major mutation along with some other VOC defining mutation as a separate column in the tables.   If they do not show temporally matched increases in sublineages, it largely alleviates this as a concern for the reader.

3. Figure 4 is very non-conventional and is hard to read.  Would suggest putting the mutations in order, listing first the gene they are from (ORF1a, ORF1b, Spike, etc), and don't include colors in the legend that aren't in the figure.  Would recommend doing away the 'protein segment' color coding and just put it in the table.

Minor issues.

Table 1 says 834 sequences, Fig 1 legend says 843. Which is it?

Plenty of typos and grammatical issues, such as on lines 105-106, 138, 207, 251, 380, and 442.

Reviewer 2 Report

This study developed a technique, iSKIM, to allow for timely identification of emergent variants.  Primarily this study used a publicly available dataset to identify intrahost variations.  Two minor variations of interest were to the Spike N501Y change and Spike L452R.  Overall, this is a good paper that compares the iSKIM methodology to standard techniques of LoFreq and ngs mapper.  The iSKIM technique is comparable to the two standard methods but appears to detect more variants sooner.  

The authors indicate that the minor variants tend to appear earlier in the population prior to becoming fixed.  Authors, should this not be expected?  Do your results using the iSKIM provide new information on how these variants arise prior to becoming fixed in the population?

Figure 2 - can you use a different color combination other than light and dark gray?  A more vibrant color for one of the data sets would be better to distinguish these results.

Also you have a large number of supplemental Tables.  Please review these Tables and supplemental Figures and determine if they are really needed to support the data in this paper.
